# Therapeutic Interventions for Music Performance Anxiety: A Systematic Review and Narrative Synthesis

**DOI:** 10.3390/bs15020138

**Published:** 2025-01-26

**Authors:** Caitlin Kinney, Phoebe Saville, Annie Heiderscheit, Hubertus Himmerich

**Affiliations:** 1Department of Psychological Medicine, Institute of Psychiatry, Psychology and Neuroscience, King’s College London, London SE5 8AF, UK; caitlinkinney11@gmail.com (C.K.); phoebedsaville@gmail.com (P.S.); 2Cambridge Institute for Music Therapy Research (CIMTR), Anglia Ruskin University, Cambridge CB1 1PT, UK; annie.heiderscheit@aru.ac.uk; 3Maudsley Hospital, South London and Maudsley NHS Foundation Trust, London SE5 8AZ, UK; 4Bundeswehr Center for Military Mental Health, Military Hospital Berlin, 10115 Berlin, Germany

**Keywords:** music performance anxiety, social anxiety disorder, intervention, therapy, treatment, music

## Abstract

The aim of this systematic review was to summarise and evaluate the published literature on interventions for treating music performance anxiety (MPA). Adhering to the PRISMA guidelines, a comprehensive literature search of three electronic databases was conducted: PubMed, Web of Science, and PsychInfo (Ovid). Records were included in this review if they were quantitative pre–post interventional studies that utilised a recognised outcome measure or a clinical diagnosis for evaluating MPA. A narrative synthesis was orchestrated on 40 extracted studies assessing 1365 total participants. The principal intervention types observed included cognitive behavioural therapy, acceptance and commitment therapy, music therapy, yoga and/or mindfulness, virtual reality, hypnotherapy, biofeedback, and multimodal therapy. Although most of the reviewed studies demonstrated encouraging improvements in musicians’ MPA following delivered interventions, the current evidence base remains in its infancy, and numerous methodological weaknesses exist across studies. Small sample sizes, heterogeneity amongst treatment programmes, lack of follow-up data, a scarcity of standardised MPA assessments, and few randomised controlled designs render it imprudent to draw definitive recommendations concerning the interventions’ efficacy.

## 1. Introduction

### 1.1. Music Performance Anxiety: Prevalence, Symptoms, and Risk Factors

Music performance anxiety (MPA) is considered to be a catalyst for poor mental health amongst professional musicians and students alike. MPA is characterised by a persistent distress and apprehension about one’s public performance skills and manifests through a distinct set of cognitive, physiological, and behavioural symptoms ([70]; [38]). MPA-related fears are deemed as markedly unwarranted when taking a pragmatic view of a musician’s ability and prior training. Nonetheless, MPA can bear a tangible impact on one’s quality of life, with severe MPA profoundly compromising a musician’s livelihood ([71]). The very experience of performing can become so aversive that, despite forcing oneself to endure it, it can no longer be tolerated. According to the [90] ([90]), persons suffering with MPA of this intensity face significant mental health consequences ([26]).

At present, performance anxiety is classified as a specifier of social anxiety disorder (SAD) under the *Diagnostic and Statistical Manual of Mental Disorders* (DSM-5; [1]). According to the [1] ([1]), an individual can present with pathological performance anxiety that meets the criteria for social anxiety disorder without fearing general social settings. The interpersonal nature of music performance and MPA-related fears of negative evaluation are closely aligned with SAD, and the relationship between the two is well documented ([19]). The prevalence of debilitating MPA amongst musicians is estimated between 16.5% and 60% ([22]), with 24% to 33% of these musicians also meeting the criteria for SAD ([21]). Severe MPA displays significant comorbidity rates with related conditions and routinely co-occurs with other anxiety disorders, depression, and substance abuse ([3]; [23]; [44]; [89]). Currently, there are no National Institute for Health and Care Excellence ([58]) guidelines for the treatment of MPA, highlighting a significant gap in the literature. Only about 15% of musicians seek support for managing their MPA, and 36.5% of music students express a need for such assistance ([35]; [87]). Accordingly, a systematic review of researched interventions for MPA could be instrumental in advancing best-practice treatment for the condition.

In terms of MPA’s clinical presentation, symptoms are frequently organised into three categories: cognitive, physiological, and behavioural ([49]; [75]). Cognitive symptoms include feelings of terror or dread, self-doubt, memory lapses, attention problems, preoccupation with failure, and hyperfixation on internal or external evaluative threats ([92]). Physiological symptoms consist of tachycardia, increased blood pressure, perspiration, shortness of breath, tremors, dizziness, and muscle tension ([87]; [91]). Behavioural symptoms often entail avoidance and safety behaviours like evading solos, compulsive practising, and resisting challenging repertoire, as well as resorting to maladaptive coping through alcohol and illicit drugs ([34]; [88]). The extent to which MPA dictates performance quality remains somewhat elusive. Performance quality can certainly falter when severe MPA is combined with inadequate preparation and a choice of repertoire surpassing one’s technical abilities; yet, in some cases, it has little effect or may even enhance performance. From a functional impairment perspective, MPA’s most detrimental consequence lies in fostering a deep-seated aversion to performance altogether. In a vicious cycle, a musician’s fear that their somatic symptoms will negatively impact a judge’s perception of their performance sparks a tridirectional feedback loop, heightening their physical symptoms, which worsens their fearful ruminations and reinforces any avoidance or maladaptive coping behaviours in place. MPA thus undermines musicians’ careers and can even push them to abandon their craft ([60]).

Certain personality characteristics emerge as potential predictive factors for MPA, namely, attributes of perfectionism, neuroticism, learned helplessness, low self-esteem, and state/trait anxiety ([4]; [43]; [84]). Early exposure to assessments in a highly competitive setting serves as an environmental risk factor increasing one’s vulnerability to MPA ([39]). Unresolved attachment ruptures, which may render audiences frightening and unsupportive, also contribute to the severity of one’s MPA ([42]). As for gender, a descriptive population study by [44] ([44]) found that women experience a greater propensity towards MPA than men, a pattern that has been replicated by subsequent research ([19]; [25]; [63]). Age emerges as another significant socio-demographic variable. Older musicians appear less impacted by MPA than their younger counterparts, with most papers reporting an adaptation to performance stressors that is gained with experience ([22]).

### 1.2. An Overview of Therapeutic Interventions for MPA

Although treatment for SAD often includes administering select anxiolytics ([58]), medication for MPA is contraindicated. Many anxiolytics impact respiratory exertion and compromise fine motor control, making them inviable for performers ([24]; [64]). Additionally, medications like beta-blockers primarily address somatic symptoms, which is marginally advantageous to MPA, given that much of its symptomology is cognitive and emotional in nature ([16]; [39]). Despite this, national survey studies have reported that upwards of 30% of professional musicians routinely take beta-blockers or benzodiazepines ([23]; [39]).

Psychotherapeutic interventions remain the principal treatments studied. Several therapeutic modalities have surfaced as promising options, including cognitive behavioural therapy (CBT), acceptance and commitment therapy (ACT), music therapy, and even hypnotherapy ([34]; [53]; [55]; [74]). Other researched approaches promote bodily awareness and relaxation for MPA, namely, yoga, mindfulness meditation, progressive muscle relaxation (PMR), and biofeedback ([46]; [48]; [50]; [82]). Certain studies also indicate the usefulness of guided imagery and virtual reality in mimicking exposure to performance settings ([5]; [20]). Moreover, eye movement desensitisation and reprocessing (EMDR) has displayed considerable potential ([9]).

### 1.3. Aims and Objectives

The present systematic review aims to synthesise and critically appraise the available literature on interventions for music performance anxiety. The efficacy of said interventions and the quality of the published evidence will be summarised accordingly.

## 2. Materials and Methods

This systematic review was conducted in alignment with the best-practice guidelines set forth in the Preferred Reporting Items for Systematic Reviews and Meta-Analyses (PRISMA) and was formulated with thoughtful adherence to the *Cochrane Handbook for Systematic Reviews of Interventions* ([62]; [30]). A pre-established research protocol served to mitigate bias and provide a methodical and transparent approach. The study procedure below is outlined in extensive detail to support its replicability in future papers. A completed PRISMA checklist is accessible in the Appendix A.

### 2.1. Search Strategy

Before commencing this review, initial scoping searches were run in December 2023, and three databases pertaining to the topic were selected: PubMed, Web of Science, and APA PsychInfo (Ovid). When developing keywords for the search strategy, the Population, Intervention, Comparison, and Outcome (PICO) framework helped to isolate core search terms, which operationalised the research question ([69]). Possible synonyms for each term were then identified using subject headings and free-text terms from the existing literature. Crucially, this cast a wide retrieval net for relevant papers and assisted in achieving a comprehensive search. Truncation symbols functioned to capture all possible variations of the root terms. The Boolean operators “OR” and “AND” connected the terms together. No filters or limits were applied to pursue the inclusiveness of each search. Full literature searches were run across the three electronic databases on the 17 July 2024, employing minor adaptations to suit each database. The following search terms were used: PubMed: (“social anxiet*”[All Fields] OR “social anxiety disorder*”[All Fields] OR “social phobia*”[All Fields] OR “performance anxiet*”[All Fields] OR “performance anxiety disorder*”[All Fields] OR “performance phobia*”[All Fields] OR “Phobia, Social”[Mesh] OR “Performance Anxiety”[Mesh]) AND (“music*”[All Fields] OR “Music”[Mesh]); Web of Science: (ALL = (“social anxiet*” OR “social anxiety disorder*” OR “social phobia*” OR “performance anxiet*” OR “performance anxiety disorder*” OR “performance phobia*”)) AND (ALL = (“music*”)); PsychInfo: (exp Social Phobia/ or “social anxiet*”.mp. or exp Social Anxiety/ or “social anxiety disorder*”.mp. or “social phobia*”.mp. or “performance anxiet*”.mp. or exp Performance Anxiety/ or “performance anxiety disorder*”.mp. or “performance phobia*”.mp.) AND (exp Music/ or music*.mp.).

### 2.2. Eligibility Criteria

The eligibility criteria detailed below were designed with [28]’s ([28]) published recommendations in mind.

#### 2.2.1. Inclusion Criteria

Participants were human.The study was a quantitative, pre–post interventional study.The study measured MPA outcomes using standardised and validated tools.Either a clinical diagnosis or a standardised questionnaire was used (e.g., the Kenny Music Performance Anxiety Inventory (K-MPAI; [38]) or the Performance Anxiety Inventory (PAI; [57]).Research was published in either English or German.

#### 2.2.2. Exclusion Criteria

Retracted articles.Case studies.Observational or naturalistic studies.Systematic reviews or meta-analyses.Graduate or doctoral theses.Perspective papers.Articles without original research.Performance anxiety trials that do not pertain to music performance.Social anxiety research that does not address music performance.

### 2.3. Study Selection and Quality Appraisal

After the electronic search was completed, all studies were aggregated into Endnote, which helped to remove duplicates. Thereafter, the studies were imported into Rayyan to detect any remaining duplicates (this process was verified by manual screening). In accordance with the PRISMA protocol, study selection involved a two-step screening process performed independently by two authors (C.K. and P.S.). In stage one, the reviewers evaluated the titles and abstracts of each study, excluding any articles that did not align with the eligibility criteria. In stage two, both reviewers then screened the articles’ full texts. During this step, the primary reviewer (C.K.) gleaned significant summary data from each article: citation details, population demographics, sample sizes, study design, outcome measures, type of intervention, results, and statistical significance. The reference lists of included studies were manually inspected through forward and backward citation tracking to identify any additional papers that fit the criteria. Using the obtained summary data, both reviewers (C.K. and P.S.) then systematically assessed each study against the appropriate Joanna Briggs Institute critical appraisal tool for either quasi-experimental research or randomised controlled trials (RCT) ([33]) (see Appendix A). Any discrepancies that arose throughout the screening process were remedied through discussion, whereupon a third independent reviewer (H.H.) was consulted in the case of unresolved disagreements.

### 2.4. Analysis

Summary data were tabulated following [66]’s ([66]) recommendations for narrative synthesis. The studies were categorised aptly into corresponding groups based on their chosen intervention.

## 3. Results

Implementing the search strategy outlined above yielded a total of 1219 records. After removing duplicates, this sum was reduced to 853. Title and abstract screening of these records produced an outstanding 80 studies, which underwent full-text screening. Upon completing full-text screening, quality appraisal was then conducted on the 40 eligible studies that remained. At this stage, none of the studies were excluded, leaving 40 papers for narrative synthesis. The PRISMA flow diagram, which documents reasons for exclusions, can be found in Figure 1. A comprehensive summary of each included study is available in Table 1. Furthermore, to provide additional clarity for the reader, a condensed synopsis of the extracted studies’ results and limitations can be found in Table 2.

### 3.1. Study Characteristics and Sample Demographics

The 40 studies included in this systematic review feature data published between 1982 and 2024, with research from the United States (*n* = 13), Australia (*n* = 7), the United Kingdom (*n* = 4), Canada (*n* = 4), Spain (*n* = 3), South Korea (*n* = 2), New Zealand (*n* = 1), South Africa (*n* = 1), Taiwan (*n* = 1), Germany (*n* = 1), Austria (*n* = 1), Nigeria (*n* = 1), and Israel (*n* = 1). As anticipated, the existing body of research predominantly focused on psychotherapeutic interventions for MPA rather than pharmacological treatments. Studies examining cognitive behavioural therapy (*n* = 6), acceptance and commitment therapy (*n* = 3), music therapy/improvisation and desensitisation (*n* = 5), yoga and/or mindfulness (*n* = 5), virtual reality (*n* = 2), hypnotherapy (*n* = 2), and biofeedback interventions (*n* = 2) comprised 62.5% of the total studies. Of the remaining studies, 22.5% implemented multimodal psychological skills programmes that incorporated various cognitive restructuring and relaxation techniques from a wide array of therapeutic frameworks (*n* = 9). The rest of the studies, 15% of the total sum, were classified under a miscellaneous category (*n* = 6).

Collectively, the included studies assessed a total of 1365 participants, 840 (65%) of whom identified as female, 450 (34.9%) as male, and 1 (0.1%) as non-binary. Two studies did not report gender-specific data. The sample sizes across studies displayed considerable variation, ranging from as few as 6 to as many as 135 (*M* = 34.13, *SD* = 26.98). Among the reviewed articles, 36 studies included participants of all genders, 3 studies exclusively featured female participants, and 1 study solely recruited male participants. One noteworthy point concerning sample demographics is that most studies did not set a minimum threshold for MPA as a criterion for participant eligibility.

Researchers routinely leveraged convenience sampling to acquire participants from local music programmes. Consequently, the majority of the studies (*n* = 36) enlisted music students as a large proportion, if not the entirety, of their sample, with 24 studies selecting only university students, 3 studies recruiting students in non-university music programmes (e.g., secondary schools), and 10 studies enlisting participants with varied musical backgrounds (e.g., students alongside teachers/community musicians/professional musicians). Three papers assessed professional musicians exclusively. Ten studies utilised participants who were uniquely specialised in a single instrument type: piano (*n* = 4), voice (*n* = 3), string (*n* = 2), and wind (*n* = 1). The participants were predominantly adults, with a distinct emphasis on young adults. Of the total papers, 62.5% (*n* = 25) had mean ages that ranged between 18 and 30, while just 15% (*n* = 6) featured participants with mean ages between 30 and 50. Five studies did not specify their participants’ ages. The remaining four papers studied adolescents or children.

### 3.2. Study Designs and Measurement Tools

Fitting with the eligibility criteria, each of the included studies was interventional in nature. Most studies employed quasi-experimental case–control designs (*n* = 26). Many studies adopted quasi-experimental one-arm designs with pre- and post-intervention assessments and no control group (*n* = 10). Similarly, several studies utilised one-arm repeated-measures designs without control groups (i.e., pre-, mid-, and post-intervention assessments) (*n* = 2). Two papers were classified as RCTs (*n* = 2). Among the 26 quasi-experimental case–control studies, the conditions used for control groups included waitlist (*n* = 11), no treatment (*n* = 8), equivalent neutral treatment (*n* = 4), active treatment (*n* = 2), and placebo (*n* = 1). Of the 40 included studies, 55% claimed that they administered random assignment to some degree (*n* = 22); however, many failed to clarify the methods used to randomise the groups. Fewer than half of the overall studies employed follow-ups (*n* = 13), meaning that 67.5% failed to do so (*n* = 27).

Remarkably, over 66 diverse questionnaires were used to measure various outcomes across studies. The most commonly adopted methods to assess MPA included psychometric self-report questionnaires, namely, the State-Trait Anxiety Inventory (STAI) ([73]), the K-MPAI ([38]), and the PAI ([57]). Physiological measures, including heart rate monitors and electromyography for muscle tension (EMG), were the next most frequently utilised measures. Ratings of performance quality and/or MPA (assessed via judged performances) were also routinely employed.

### 3.3. Study Quality

Taken together, the reviewed studies had a moderate risk of bias, with less than 40% negative or unclear responses per quality appraisal tool. For all quasi-experimental studies (*n* = 38), it was evident that suitable statistical analysis was used, outcomes were generally measured in a reliable manner, the intervention and outcome variables were explicitly stated, and there were multiple measurements of MPA (pre- and post-intervention). Nevertheless, 13 of these studies did not administer control groups and, thus, failed to provide the same calibre of causal evidence as those with comparators. Among the studies with a case–control design (*n* = 25), outcomes were measured similarly across all groups. All participants were typically recruited from the same music programmes (ensuring consistent socio-demographic characteristics across groups). Only two studies implemented active treatment control groups that were concerningly dissimilar to their chosen intervention (potentially introducing confounders). Of the 13 studies with follow-ups, data were complete in eight papers. Attrition was not suitably analysed and described in three studies.

For both RCTs (*n* = 2), the treatment groups were similar at baseline and were treated identically other than the intervention of interest, follow-up was complete, true randomisation was used, outcomes were measured reliably, and the appropriate trial methodology and statistical analyses were executed. However, the nature of the interventions made it difficult for the participants or treatment administrators to be blinded to the group assignments. It was unclear whether group allocation was adequately concealed.

**Figure 1 behavsci-15-00138-f001:**
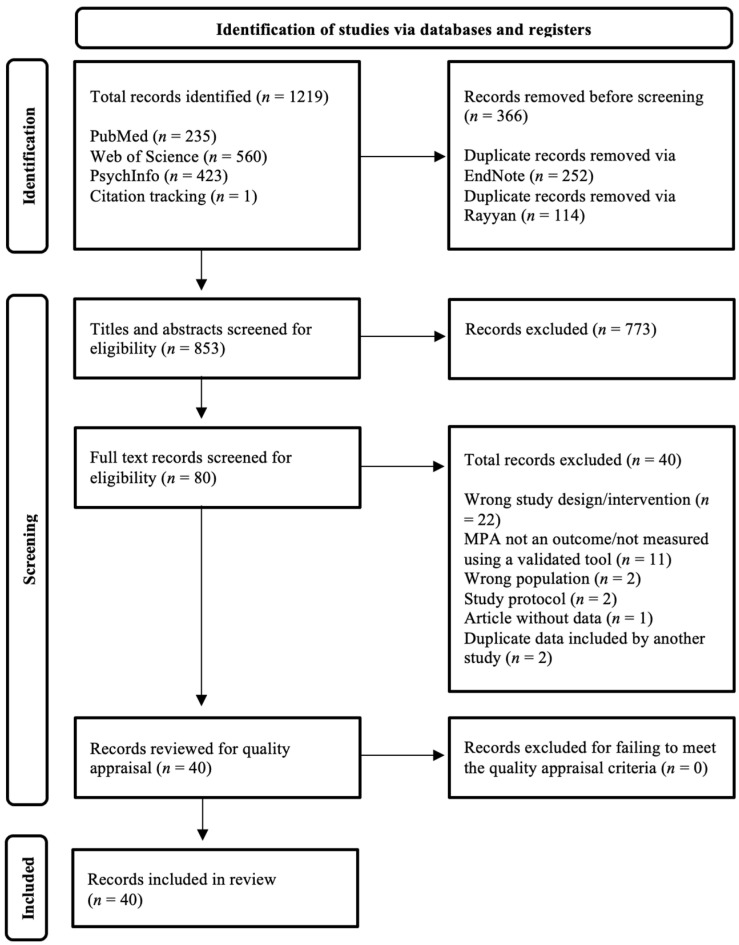
PRISMA flow diagram.Abbreviation: Music Performance Anxiety (MPA).

**Table 1 behavsci-15-00138-t001:** Comprehensive summary of included studies.

Authors/Year/Country	Sample	Age	Study Design	Control Group	Measures	Type of Intervention	Key Outcomes and Statistical Significance
**(1) Cognitive Behavioural Therapy Interventions for Music Performance Anxiety**							
**[15] ([15]),** **United States**	Individuals who reported that MPA was impairing their life and met DSM criteria for social phobia. Total (*n* = 34). Three intervention groups: CBT with placebo (*n* = 9), CBT with buspirone (*n* = 8), buspirone alone (*n* = 9). Control: Placebo alone (*n* = 8).	*M* = 37.5 Gender not provided	Quasi-experimental study (pre–post- test with a one-month follow-up) with randomised groups. No allocation concealment. Method of randomisation unclear.	Placebo control group (6 weeks of placebo)	PRCP; Self-statement questionnaire; SADS. Fear of negative evaluation scale; Heart rate monitor.	- CBT with placebo group: One session of group CBT per week for 5 weeks. Sessions helped participants identify cognitive distortions, develop coping models, and provided graded exposure to feared performance situations. - CBT with buspirone group: The same CBT programme as above, alongside a starting dose of 5 mg tablets of buspirone three times per day (with dosing increased to maximum tolerated amounts). - Buspirone alone group: starting dose of 5 mg tablets of buspirone three times per day (with dosing increased to maximum tolerated amounts), 6 weeks.	CBT (with placebo) yielded significant improvements in subjective anxiety (*p* < 0.05), musical performance quality (*p* < 0.05), and performance confidence (PRCP, *p* < 0.01) compared to other treatments. Interestingly, participants taking buspirone alone did not experience any statistically significant changes in their subjective anxiety or performance quality compared to participants taking the placebo. Furthermore, participants taking buspirone reported dizziness more frequently as a side effect (*p* < 0.001).
**[36] ([36]),** **Canada**	Music students of all ages identified by their teachers as having extreme MPA; Total (*n* = 53); Male (*n* = 5); Female (*n* = 48). Two intervention groups: attentional training (*n* = 19) and behaviour rehearsal (*n* = 16). Control (*n* = 18).	12–53 *M* = 18.83	Quasi-experimental study with randomised assignment to two treatment groups or one control. No allocation concealment. Method of randomisation unclear. Pre- and post-test performances; 5-week follow-up.	Waitlist control group	SAI; SSS; EPES; PASSS; heart rate; performance error counts and MPA behavioural reports were recorded by raters.	A 3-week programme with one 1.5–2 h group session per week, alongside homework. - Attentional training: CBT sessions focusing on recognising and substituting negative thoughts with positive, task-oriented self-statements. Participants engaged in cognitive modelling, visualisation of examination and recital performances, and performing in front of family to challenge negative self-talk. - Behaviour rehearsal: Repeated performance practice before a supportive audience with positive feedback. Sessions aimed to reduce MPA through exposure and included education about the origins of MPA.	CBT (attentional training) and behaviour rehearsal were both effective in decreasing MPA compared to the control group at follow-up (*p* < 0.01), although these differences were not visible immediately after treatment and only emerged at the 5-week follow-up. The CBT (attentional training) group showed superior reductions in visual signs of anxiety (*p* < 0.05) and enhancements in self-efficacy (*p* < 0.05) relative to behaviour rehearsal alone.
**[45] ([45]),** **Australia**	Secondary school students, undergraduate and postgraduate music students, and community musicians. Total (*n* = 68); Male (*n* = 23); Female (*n* = 45).	16–81 *M* = 44.51	Quasi-experimental design (pre- and post-tests). Randomised groups. No allocation concealment. Method of randomisation unclear. All participants acted as the intervention and control group. Four performances: baseline, after the placebo, after the intervention, and at 4–6-week follow-up.	Pedagogical practice skills session	K-MPAI; STAI; ASI; PRIME-MD-PHQ; judges’ performance quality assessment.	Participants were allocated into one of two brief group interventions. Both were delivered in one day over five hours: - CBT-based intervention: psychoeducation, goal-setting, cognitive restructuring, emotion surfing, and exposure tasks. - Anxiety sensitivity (AS) reduction intervention: psychoeducation, breathing retraining, muscle relaxation, interoceptive exposure techniques.	Decreases in state anxiety were observed in the AS reduction group between the second performance (post-placebo) and the third performance (post-intervention) (*p* = 0.04). This effect was not statistically significant for the CBT group. Both intervention groups demonstrated noteworthy improvements in performance quality across all performances (*p* = 0.001).
**[56] ([56]),** **United States**	Undergraduate music students experiencing debilitating MPA. Total (*n* = 20); Male (*n* = 8); Female (*n* = 12); Intervention (*n* = 12); Control group (*n* = 8)	Age not provided	Quasi-experimental interventional pilot study (pre- and post-test without follow-up) with randomised groups. No allocation concealment. Method of randomisation unclear.	Waitlist control group	PAI; STAI; TAI; APQ; RBI.	A 6-week programme combining CBT, progressive muscle relaxation (PMR), and temperature biofeedback training. Sessions included substituting negative thoughts with positive coping statements, identifying/challenging irrational thought patterns, MPA-related psychoeducation, and cognitive restructuring. A group CBT and PMR class and an individual temperature biofeedback training class occurred once per week.	The treatment group exhibited a marked reduction in performance anxiety compared to the control group (PAI, *p* < 0.001), thus confirming the effectiveness of the treatment. There were also notable improvements in trait anxiety compared to the control group (STAI-T, *p* < 0.02). Differences in state anxiety and physiological measures were less pronounced.
**[61] ([61]),** **Australia**	Adolescent music students from a selective music school with the top 25% of MPAI-A scores. Total (*n* = 23); Male (*n* = 9); Female (*n* = 14); Intervention (*n* = 14); Control (*n* = 9).	*M* = 13.87	Quasi-experimental study with non-randomised pre-selection followed by random assignment. No allocation concealment. Method of randomisation unclear. Pre- and post-test without a follow-up. MPA was measured at two performances: one pre- and one post-treatment.	Behaviour-exposure-only control group	STAI; MPAI-A; heart rate monitor; EMG to measure muscle tension.	A 7-session CBT programme designed for secondary students with a combination of individual and group sessions. Topics included awareness, goal-setting, relaxation techniques, cognitive restructuring, self-talk, performance routines, and practices for thinking realistically. - Weekly sessions (1 h for group, 45 min for individuals).	Students who were adherent to the CBT programme (i.e., students who were actively engaged) showed significant improvements in MPA (MPAI-A scores) compared to non-adherent students and the control group (*p* = 0.000). These students also had higher levels of baseline anxiety (*p* > 0.02). There appeared to be no significant effects of CBT on performance quality, STAI, or heart rate (*p* > 0.05).
**[81] ([81]), New** **Zealand**	Conservatory music students. Total (*n* = 14); Male (*n* = 8); Female (*n* = 6).	Age not provided	One-arm interventional study (pre- and post-test design with 4-month follow-up).	N/A	M-PAS; STAI; EPQ-R.	A CBT-based therapy consisting of psychoeducation about anxiety, examination of performance history, replacing negative thoughts with positive ones, attention techniques, pre-performance routines, and personality trait examination. Each session included group discussion and setting personal application exercises as homework. - One weekly 1 h group session for 6 weeks.	Music students experienced a significant reduction in their MPA levels (M-PAS scores) from pre-test to post-test to four-month follow-up (*p* = 0.03).
**(2) Acceptance and Commitment Therapy Interventions for Music Performance Anxiety**							
**[16] ([16]),** **Australia**	Undergraduate student vocalists. Total (*n* = 6); Male (*n* = 2); Female (*n* = 4).	*M* = 20.33	One-arm interventional study (pre- and post-test design with a 3-month follow-up).	N/A	K-MPAI; MPFI; DASS-21; MHC-SF.	Group acceptance and commitment therapy (ACT) programme with practices including mindfulness, acceptance strategies, cognitive defusion, and value-based activities. - One 90 min session for 8 weeks.	Students experienced notable decreases in MPA (*p* < 0.019) and psychological inflexibility (*p* = 0.026), which were maintained at 3-month follow-up. Psychological flexibility scores increased significantly post-intervention (*p* < 0.05) and were sustained at follow-up. Overall wellbeing levels showed pronounced improvement (*p* = 0.03), but this effect was not seen at follow-up. Depression, anxiety, and stress changes were not statistically significant.
**[34] ([34]),** **United States**	Undergraduate and postgraduate student vocalists with self-reported MPA. Total (*n* = 7); Male (*n* = 1); Female (*n* = 6).	19–31 *M* = 23.29	One-arm interventional pilot study (pre- and post-test design with follow-ups at 1 and 3 months).	N/A	K-MPAI; ACQ; PHLMS; AAQ-2; BAFT; VLQ; EES; MPQ.	A 12-week acceptance and commitment therapy (ACT) programme consisting of an orientation to ACT, developing acceptance for anxiety-related discomfort, mindfulness and meditation training, defusion techniques, value identification, exposure exercises, in-session performances, and homework.	Participants experienced significant reductions in MPA (K-MPAI) and anxiety control (ACQ) scores post-treatment, which were maintained at both follow-ups (*p* < 0.05). Improvements were also observed in participants’ cognitive defusion (BAFT) and acceptance/psychological flexibility (AAQ-2) scores at post-treatment and follow-up (*p* < 0.05). Performance quality showed notable benefits reflected through judges’ ratings (*p* < 0.05).
**[51] ([51]),** **United Kingdom**	Undergraduate musical theatre and dance students. Total (*n* = 6); Male (*n* = 2); Female (*n* = 4).	18–20 *M* = 19.33	One-arm mixed-methods repeated-measures design (with pre-, mid-, and post-coaching tests, as well as a 3-month follow-up).	N/A	K-MPAI; ESS; SCS; AAQ-2; BAFT; PHLMS; interviews.	Acceptance and commitment (ACT) coaching consisting of mindfulness, acceptance of MPA symptoms, defusion from MPA-related thoughts, value identification, self-compassion exercises, and psychological flexibility practices. Online. - One 1 h group session for 6 weeks.	Students displayed significant reductions in MPA, fusion of MPA-related thoughts, and state shame during performance, which were maintained at follow-up (all *p* < 0.05). Improvements were also observed in psychological flexibility, mindful acceptance, and social comparison, similarly maintained at follow-up (all *p* < 0.05). Performance quality ratings showed no significant benefits.
**(3) Music Therapy Interventions for Music Performance Anxiety**							
**[17] ([17]), Canada**	Undergraduate music students. Total (*n* = 15); Male (*n* = 1); Female (*n* = 13); Non-binary (*n* = 1).	21–33 *M* = 24.4	One-arm explanatory sequential mixed-methods study (pre- and post-test design and no follow-up).	N/A	STAI; Likert scale ratings of anxiety and mood; questionnaires; interviews.	Music psychotherapy incorporating breathing exercises, guided musical relaxation, visualisation, grounding techniques, and musical improvisation. In-person or online. - One 1 h group session for 6 weeks.	Students did not exhibit any significant differences in state or trait STAI anxiety scores (*p* > 0.05). Likert scale reports indicated improvement in mood and anxiety post-intervention (*p* < 0.001).
**[47] ([47]),** **United States**	Undergraduate and postgraduate pianist music students. Total (*n* = 6); Female (*n* = 6). Participants served as their own controls.	*M* = 25	One-arm interventional study (pre- and post-test design without a follow-up).	N/A	STAI-S; STAI-T; PARQ; LAS.	Music therapy improvisation and desensitisation protocol (MTIDP), which included rhythmic breathing exercises, free improvisation, meditation, guided imagery, and systematic desensitisation training. - One weekly 30 min one-on-one training session with a music therapist for 6 weeks.	Considerable reductions in anxiety were observed across both the LAS (*p* = 0.024) and STAI-S (state) scales (*p* = 0.028). No significant trends were seen across either the STAI-T (trait) scale or the PARQ.
**[48] ([48]),** **South** **Korea**	University pianist students. Total (*n* = 30); Female (*n* = 30). Two active treatment groups, each with *n* = 15.	*M* = 20	Quasi-experimental study (pre- and post-test without follow-up) with two active randomised treatment groups. No allocation concealment. Method of randomisation unclear. Participants performed two lab recitals (before and after therapy).	N/A	VAS; STAI-S; MPAQ; finger temperature.	A 6-week music therapy programme according to their assigned group. One 30 min session per week. - Group 1: Improvisation-assisted desensitisation. Rhythmic breathing, free improvisation, guided meditation, desensitisation training, and homework assignments. - Group 2: Music-assisted progressive muscle relaxation (PMR) and imagery. Muscle relaxation exercises while listening to preferred music, guided meditation, rhythmic breathing, and homework assignments.	The music-assisted PMR and imagery group showed statistically significant improvements in MPA (MPAQ, *p* = 0.002), state anxiety (STAI-S, *p* = 0.004), tension (*p* = 0.000), comfort (*p* = 0.03), and finger temperature (*p* = 0.001). The improvisation-assisted desensitisation group showed statistically significant reductions in state anxiety (*p* = 0.033), tension (*p* = 0.016), and finger temperature (*p* = 0.014), but not MPA. Overall, no statistically significant differences emerged between the two groups for these measures (*p* > 0.05).
**[53] ([53]), United States**	Freelance musicians who self-identified as overly anxious. Total (*n* = 17); Male (*n* = 8); Female (*n* = 9); Intervention (*n* = 7); Control (*n* = 10).	18–48 *M* = 20.9	Quasi-experimental study (pre- and post-test without follow-up) with randomised groups. No allocation concealment. Method of randomisation unclear.	Waitlist control group	STAI-T; PRCP.	Group music therapy consisting of 12 weekly 1.5 h sessions. The programme included warm-up relaxation and breathing exercises, group musical improvisation, group discussion, role-playing, guided imagery, and music-related self-statements. Participants performed the same piece for the group and outside audience members at 3 points throughout the programme.	Group music therapy significantly reduced musicians’ trait anxiety (STAI-T, *p* < 0.013) and improved performance confidence (PRCP, *p* < 0.009) compared to the waitlist control.
**[68] ([68]), United States**	Undergraduate and postgraduate music students. Total (*n* = 18); Male (*n* = 5); Female (*n* = 13). Two intervention groups: music-assisted group (*n* = 6) and verbal-only group (*n* = 6); Control (*n* = 6).	19–45 *M* = 26.66	Quasi-experimental study (pre- and post-test without follow-up) with two active randomised treatment groups. No allocation concealment. Method of randomisation unclear. Both groups completed a performance in front of an audience and judges pre- and post-test.	Waitlist control group	STAI; PARQ; MPAQ; heart rate (HR) monitor; EMG to measure muscle tension.	Both intervention groups underwent 8 weekly group sessions of coping with systematic desensitisation, progressive relaxation training, and arranging fear-arousing stimuli hierarchically in small steps to aid the desensitisation process. Each session lasted 75 min. The intervention also incorporated home practice. In contrast to the verbal group, the music-assisted group listened to their preferred relaxation music during in-person and at-home sessions.	No significant differences were found between groups in any of the physiological MPA or self-reported anxiety measures administered in the study (*p* > 0.05).
**(4) Yoga and/or Mindfulness Interventions for Music Performance Anxiety**							
**[11] ([11]), United States**	Young adult musicians in summer fellowship programmes. Total (*n* = 103); Male (*n* = 46); Female (*n* = 57); Intervention (*n* = 60); Control (*n* = 43).	Yoga: *M* = 25.2 Control: *M* = 23.6	Quasi-experimental study of data over 3 years (2005, 2006, 2007). Pre- and post-test measurements without follow-up. Non-randomised groups.	Untreated control group	DSF-2; FFMQ; POMS; PAQ.	An 8-week Kripalu yoga programme with slight variations across three years. - 2005: A day-long intensive orientation including 2 yoga practice sessions, learning about yoga theory, and an intro to meditation. Daily yoga sessions of varying intensity (students could determine their own schedule). One weekly 1.5 h yoga class with a 2 h group discussion. Optional 30 min meditation sessions (5 days a week). An end-programme retreat. - 2006: Like above, but participants were randomised into two groups (yoga-only vs. yoga lifestyle), wherein yoga lifestyle also included a 2-day retreat, one mandatory 1 h private yoga session per week, and one 1 h problem-solving discussion per week. - 2007: Like years prior, with a 1-day introductory retreat, one 1 h yoga session and 1 h problem-solving discussion per week, and three 1.25 h yoga and/or meditation sessions per week. Optional 30 min meditation sessions (5 days a week).	Total dispositional flow (DFS-2) significantly improved following yoga compared with participants in the control group (*p* < 0.05). The 2006 cohort showed significant improvements in the mindfulness subscale of awareness (*p* < 0.01). However, this was not consistently observed in 2007. Compared to controls, yoga participants’ increases in dispositional flow and mindfulness scores were associated with significant decreases in MPA (PAQ) in practice (*p* < 0.01, *r* = -0.40; *p* < 0.05, *r* = -0.26, respectively), group (*p* < 0.01, *r* = -0.44; *p* < 0.01, *r* = -0.35, respectively), and solo performance contexts (*p* < 0.01, *r* = -0.51; *p* < 0.01, *r* = -0.36, respectively).
**[13] ([13]), United States**	Undergraduate and postgraduate music students. Total (*n* = 19); Male (*n* = 5); Female (*n* = 14); Intervention (*n* = 9); Control (*n* = 10).	18–41 *M* = 25.1	Quasi-experimental study (pre- and post-test without a follow-up) with randomised groups. Both groups completed a public performance post-test. No allocation concealment. Method of randomisation unclear.	Waitlist control group	PAI; STAI-S; CIQ.	A meditation programme focused on a variety of meditation types (sitting, standing, laying down, sleeping), as well as mindfulness, relaxation, direct contemplation, awareness of the body, and stretching. Participants were also instructed to meditate for 20 min daily, especially before music practice sessions. - One weekly 1 h and 15 min session for 8 weeks.	The meditation group experienced a statistically significant decline in MPA (PAI scores) from pre- to post-performance testing compared to the control group (*p* < 0.05). Differences in post-test state anxiety (STAI-S) and cognitive interference (CIQ) were not significant, despite having moderate effect sizes (*d* = 0.5).
**[46] ([46]), United States**	Young adult musicians in a summer residential programme, Total (*n* = 135); Male (*n* = 59); Female (*n* = 76); Intervention (*n* = 84); Control (*n* = 51).	*M* = 16.4	Quasi-experimental (pre- and post-tests without follow-up) with non-randomised groups.	Untreated control group	PAQ; MPAI-A; STAI; PRMD-Q.	A Kripalu yoga programme incorporating physical postures, breathing techniques, and meditation practices aimed at alleviating performance-related musculoskeletal pain and improving body awareness. Kripalu yoga is considered “a meditation in motion”. - Three weekly 1 h group sessions for 6 weeks.	Yoga programme participants experienced significant reductions in MPA compared to controls (MPAI-A, *p* < 0.001; PAQ, *p* < 0.05 in group performance contexts; PAQ, *p* < 0.01 in solo performance contexts). No significant differences in state anxiety were observed overall (STAI, *p* > 0.05). Findings regarding performance-related musculoskeletal disorders (PRMDs) were inconsistent.
**[50] ([50]), United States**	Undergraduate and postgraduate music conservatory students. Total (*n* = 19); Male (*n* = 5); Female (*n* = 14); Intervention (*n* = 9); Control (*n* = 10).	19–41 *M* = 25.1	Quasi-experimental study with a randomised between-group design (pre- and post-test without follow-up). No allocation concealment.	Waitlist control group	SAI; PAI; MPQ; visual analogue scale of MPA from 0 (low anxiety) to 15 (high anxiety).	A Chan (Zen) group meditation programme that incorporated various modalities of meditation as well as bodily awareness, mental rehearsal, and performance visualisation. Participants also practiced for 20 min daily at home, especially before practicing music. - One 1 h and 15 min group session for 8 weeks.	Meditation programme participants did not experience any marked enhancements in performance quality (*p* = 0.32), although a positive correlation was found between performance quality and MPA in the meditation group (*r* = 0.682, *p* = 0.043). In contrast, a negative correlation emerged between performance quality and anxiety (both MPA, *r* = -0.711, *p* = 0.021 and state anxiety *r* = -0.718, *p* = 0.019) in the control group.
**[76] ([76]), United States**	Undergraduate and postgraduate music conservatory students. Total (*n* = 24); Male (*n* = 3); Female (*n* = 21).	18–29	One-arm interventional pilot study (pre- and post-test design with a 7–14-month follow-up).	N/A	PAQ; K-MPAI; STAI-T; POMS.	A Kripalu yoga programme consisting of physical postures, breathing techniques, and meditation practices. Students engaged with two 1 h group yoga sessions for 9 weeks. The programme also included home practice for 4 days a week through 16 min CDs. Optional brief spontaneous breathing techniques were encouraged when participants noticed that they were becoming stressed.	Large decreases were observed in trait anxiety (STAI, *p* = 0.001, *d* = 1.05) and solo music performance anxiety scores (PAQ, *p* = 0.002, *d* = 0.99). K-MPAI scores decreased significantly from baseline to post-intervention (*p* = 0.003, *d* = 0.87). Improvements were sustained at the 7–14-month follow-up.
**(5) Virtual Reality Interventions for Music Performance Anxiety**							
**[5] ([5]), Canada**	Post-secondary, college, and university, and conservatory music students. Total (*n* = 17); Male (*n* = 7); Female (*n* = 10); Intervention (*n* = 9); Control (*n* = 8).	*M* = 21.8	Quasi-experimental (pre- and post-tests, without follow-up) with randomised groups. Before and after training, participants played musical pieces in a public recital. No allocation concealment. Method of randomisation unclear.	Untreated control group	STAI-S; PAI; PRCP; quality of performances rated by 2 judges.	Virtual reality exposure training (VRET) that subjected participants to different virtual environments simulating classical performance situations (including audiences and judges). Psychoeducation on anxiety was an element of the first session. - Six 1 h private sessions for 3 weeks.	Significant reductions in MPA (PRCP scores) were seen for the VRET group (*p* < 0.05), but not for controls. Likewise, significant increases in performance quality emerged for the VRET participants (*p* < 0.05), but not for controls. Significant decreases in state anxiety (STAI-S) were present for those in the VRET group with high anxiety at pre-test (*p* < 0.01), but not for those in the VRET group with low state anxiety at pre-test, or for controls.
**[14] ([14]), South** **Korea**	Professional and student vocalists. Total (*n* = 18); Male (*n* = 13); Female (*n* = 5).	21–37	One-arm interventional study (pre- and post-test design, and no follow-up).	N/A	K-MPAI; SADS; pulse monitor.	Virtual reality exposure therapy in 4 settings: stage performance, studio recording, closed audition, and open audition. Each participant underwent 2 of 4 performance sessions in person.	Participants showed substantial reductions in all dimensions of MPA (affective, cognitive, relational, behavioural) after the first exposure session (*p* < 0.05), but not after the second exposure session (*p* < 0.05).
**(6) Hypnotherapy Interventions for Music Performance Anxiety**							
**[9] ([9]), United Kingdom**	Postgraduate and undergraduate pianists suffering from MPA. Total (*n* = 46); Male (*n* = 17); Female (*n* = 29).	18–53	Quasi-experimental study (pre- and post-tests, and no follow-up) with randomised groups. No allocation concealment. Method of randomisation unclear.	Untreated control group	STAI; self-report questionnaire; Likert scale for performance assessment.	Two therapeutic sessions over two weeks of either cognitive hypnotherapy (CH) or eye movement desensitisation and reprocessing (EMDR), each lasting 1 h.	Both CH and EMDR therapy groups (but not the control group) experienced significant reductions in state anxiety post-treatment (CH: *p* = 0.005; EMDR: *p* = 0.017) and a significant improvement in performance (CH: *p* = 0.012; EMDR: *p* = 0.022). The EMDR group exhibited notably lowered trait anxiety compared to the control group (*p* = 0.005) and the CH group (*p* = 0.015).
**[74] ([74]), Australia**	Second- and third-year university music students identified by their teachers as being prone to MPA. Total (*n* = 40); Intervention (*n* = 20); Control (*n* = 20).	Age and gender not specified	Quasi-experimental study with measures taken pre-, post-, and 6 months after the intervention. Pairs of students were matched based on their PAI scores and then randomly assigned to a group. No allocation concealment. Method of randomisation unclear.	Controls had two 50 min sessions discussing stress and coping strategies	PAI.	A hypnotherapy intervention combining relaxation techniques (hypnotic breathing), symbolic mental imagery, and verbal suggestions linking the chosen symbols to beliefs of confidence, calmness, and increased mental control. - Two 50 min sessions (each a week apart).	Hypnotherapy effectively reduced MPA in students, as indicated by a significant drop in PAI scores from pre-intervention to post-intervention and 6 months post-intervention (*p* < 0.01). By contrast, the control group showed no substantial improvement.
**(7) Physiological and/or Biofeedback Interventions for Music Performance Anxiety**							
**[82] ([82]), United States**	Undergraduate and postgraduate music students. Total (*n* = 14); Male (*n* = 9); Female (*n* = 5); Intervention (*n* = 7); Control (*n* = 7).	19–32 *M* = 23	Quasi-experimental study with randomised groups. Pre- and post-tests without follow-up. No allocation concealment. Non-rigorous randomisation.	Untreated control group	STAI; PAI; FSS; heart rate variability (HRV).	Treatment included heart rate variability (HRV) coherence biofeedback training and emotional regulation techniques. The intervention included learning about physiological arousal, the heart–brain connection, emotional memory, and other elements of psychoeducation. Students were trained to improve their HRV through relaxed breathing and emotional refocusing techniques. - One 30–50 min session for 4–5 weeks.	The treatment group displayed statistically significant improvements in combined anxiety scores (i.e., combining the PAI with the mental/emotional components of the STAI and physiological HRV), with a large effect size (*p* < 0.05, ηp² = 0.320). HRV also showed statistical significance with a large effect size (*p* = 0.01, ηp^2^ = 0.698). In contrast, state anxiety (STAI, *p* = 0.057, ηp^2^ = 0.291), performance anxiety (PAI, *p* = 0.192, ηp^2^ = 0.149), and average heart rate (*p* = 0.203, ηp^2^ = 0.143) did not reach conventional significance thresholds, despite all having large effect sizes.
**[86] ([86]), Australia**	Musicians of all ages. Total (*n* = 44); Male (*n* = 20); Female (*n* = 24). Intervention: breathing with biofeedback (*n* = 14), breathing only (*n* = 15). Control (*n* = 15).	19–67 *M* = 30.4	Randomised controlled trial with a 3 (group) x 2 (time) mixed experimental design (pre- and post-tests without follow-up).	Untreated control group was instructed to read during intervention period	STAI-S; heart rate variability (HRV): high frequency (HF), LF/HF ratio.	A single 30 min session of slow breathing (with or without biofeedback). Participants received diaphragmatic breathing instruction, followed by exercises using a breathing pacer at 6 breaths per minute. The breathing with biofeedback group received feedback from the pacer, while the breathing-only group did not.	During performance anticipation, slow breathing groups (with or without biofeedback) showed significant improvements in HRV (increased HF, decreased LF/HF ratio) compared to the control group (*p* = 0.026 and *p* = 0.017, respectively). Initially highly anxious individuals displayed significant reductions in state anxiety (STAI-S) after the interventions (*p* < 0.05), while no such effects were observed in those with low state anxiety before the intervention. The addition of biofeedback did not produce differential results.
**(8) Multimodal Interventions for Music Performance Anxiety**							
**[7] ([7]), Australia**	Secondary school music students. Total (*n* = 62); Female (*n* = 62); Intervention (*n* = 30); Control (*n* = 32).	*M* = 13.8	Quasi-experimental study with pseudo-randomised groups. Pre- and post-tests, with 2-month follow-up. Before and after training, participants played musical pieces in a judged recital.	Waitlist control group	MPAI-A; judge-rated scales for MPA and performance quality.	An 8-week “Unleash Your Potential” intervention involving weekly group sessions led by school psychologists. Each session focused on topics like goal-setting, identification of strengths, relaxation techniques, cognitive restructuring, mental imagery, visualisation, stress management, focus/flow exercises, and processing failures/developing resilience.	Significant reductions in MPA (MPAI-A scores) were found in both groups after the intervention compared to their baseline (*p* < 0.001), and this reduction was maintained at the 2-month follow-up. These improvements did not translate into judge-rated MPA (*p* > 0.05) or judge-rated performance quality (*p* > 0.05).
**[8] ([8]), United Kingdom**	Professional symphony orchestra musicians. Total (*n* = 54); Male (*n* = 27); Female (*n* = 27). Intervention (*n* = 18 per group): music therapy, vibrotactile, standard counselling.	22–55 *M* = 36	Quasi-experimental study with randomised assignment to one of four groups (three treatment groups and one control). Two-month follow-up. No allocation concealment. Method of randomisation unclear.	Traditional counselling control group	STAI-T; AMPS; MPSS; DSP; POMS; MBI.	An 8-week programme with one weekly 1 h session. Assigned to one of three groups: - Counselling group: verbal counselling, progressive muscle relaxation exercises, and CBT techniques. - Music group: verbal counselling, progressive muscle relaxation exercises, and CBT techniques, supplemented with music - Vibrotactile group: verbal counselling, progressive muscle relaxation exercises, and music-generated vibrotactile sensations (somatron acoustic massage power recliner).	Substantial therapeutic gains emerged across all groups, yet all programmes (music, vibrotactile, and standard counselling) displayed similar levels of effectiveness. Key results included significant reductions in trait anxiety (STAI-T, *p* = 0.004), performer’s stress (AMPS, *p* = 0.001), and anxiety/depression/confusion/mood disturbance (POMS, *p* < 0.001). These effects were sustained at follow-up.
**[18] ([18]),** **Israel**	Post-graduate music therapy students. Total (*n* = 24); Male (*n* = 5); Female (*n* = 19); Intervention (*n* = 12); Control (*n* = 12).	23–45 *M* = 30.54	Quasi-experimental with pre- and post-tests; no follow-up; non-randomised groups.	Waitlist control group	PAI; STAI; PANAS; DFS-2; BSI; PQ; SPA.	Music performance skills course comprising mental skills training (positive thinking, goal setting, mental rehearsal, negative automatic thought identification, etc.), physiological awareness (regulating arousal, centring techniques), simulated performances, and enhancing musical communication (improvisation). - One 90 min group session for 11 weeks.	The intervention group displayed substantial declines in MPA compared to waitlist controls (*p* < 0.001). For the intervention group, significant differences in state anxiety (*p* < 0.05) and negative affect (*p* < 0.05) were seen before the second performance. After the second performance, improvements in both positive affect (*p* < 0.05) and negative affect (*p* < 0.01) were observed.
**[27] ([27]), Spain**	Postgraduate classical music performance students. Total (*n* = 17); Male (*n* = 9). Female (*n* = 8).	22–26	One-arm interventional study (pre- and post-test design and a 1-year follow-up).	N/A	K-MPAI; qualitative satisfaction questionnaire; interviews.	“ConfiDance” programme combining reflection and reprocessing techniques from CBT and ACT, visualisation, body awareness, mindfulness, yoga, music creation, playful performance, and expressive writing. In person. - One 2 h group session for 8 weeks.	Music students experienced reductions in MPA, with long-term effects seen at follow-up (*p* < 0.05). Women exhibited higher MPA than men (40% of women exceeded a debilitating MPA threshold, compared to 26.66% of men).
**[31] ([31]), Australia**	Student, amateur, and professional musicians. Total (*n* = 33); Male (*n* = 4); Female (*n* = 29); Intervention (*n* = 15); Control (*n* = 18).	19–66 *M* = 42.09	Quasi-experimental study with a 2 (group) × 2 (time) repeated-measures design and randomised groups. One month follow-up. No allocation concealment. Method of randomisation unclear.	Waitlist control group	STAI; PAI; BAI; heart rate monitor; performance quality ratings by judges.	A cognitive restructuring and imagery intervention with workshops covering topics like managing performance-related thoughts, self-awareness exercises, using self-talk, visualising successful performances, and identifying/amending dysfunctional thought patterns. - One 1 h group session for 3 weeks.	The intervention effectively reduced MPA (PAI, *p* < 0.05) and improved judge-rated performance quality (*p* < 0.05) in participants from pre- to post-test. Control groups showed no such improvements. Benefits to anxiety were maintained at a one-month follow-up. Nonetheless, there were no significant changes in state anxiety (STAI, *p* = 0.25) or physiological measures like heart rate (*p* = 0.60).
**[54] ([54]), Spain**	Music conservatory students and active performing music teachers. Total (*n* = 62); Male (*n* = 20); Female (*n* = 42); Intervention (*n* = 28); Control (*n* = 34).	18–61 *M* = 27.58	Quasi-experimental non-equivalent comparison group design (pre- and post-tests and no follow-up); non-randomised groups.	Untreated control group	K-MPAI; EFIM; SSS.	“Self-regulation Skills for Performing Musicians” programme covering emotional and social awareness, mindfulness, performance preparation exercises, and regulation techniques. Based on flow theory and positive psychology. Online; 12 weeks.	The intervention group demonstrated a marked improvement in MPA (*p* = 0.01) and flow state scores (*p* = 0.02) compared to the control group. There were no observed differences in social skills.
**[59] ([59]),** **Nigeria**	Undergraduate music students. Total (*n* = 70); Male (*n* = 31); Female (*n* = 39); Intervention (*n* = 35); Control (*n* = 35).	17–23	Randomised controlled trial with pre- and post-tests and 2-week follow-up.	Waitlist control group	K-MPAI; PSS.	Educational music training including active participation in preferred music using percussion instruments, alongside stretching, breathing, and discussing themes of anxiety. - Two weekly 40 min sessions for 8 weeks.	Students who received educational music training experienced substantially reduced MPA levels (*p* = 0.001) and perceived stress (*p* = 0.001) compared to students in the control group. These improvements were sustained at follow-up.
**[72] ([72]), Germany**	Undergraduate student string players. Total (*n* = 21); Male (*n* = 7); Female (*n* = 14); Intervention (*n* = 13); Control (*n* = 8).	*M* = 22.1	Quasi-experimental (pre- and post-tests without follow-up), with non-randomised groups and simulated performances.	Waitlist control group	STAI-S; K-MPAI; FZAQ.	A 14-week multimodal intervention programme designed to train coping strategies for MPA, with weekly 90 min sessions. The programme involved autogenic training, breathing exercises, body awareness, imaginative techniques, cognitive strategies, video feedback, and practice performances.	The intervention group displayed marked reductions in state anxiety (STAI-S) from the first performance (pre-test) to the second performance (post-test) compared to controls (*p* < 0.01), alongside significant enhancements in performance as rated by judges (*p* < 0.05).
**[77] ([77]), South** **Africa**	Undergraduate music students. Total (*n* = 36); Male (*n* = 15); Female (*n* = 21). Intervention/control samples not provided.	Age not provided	Quasi-experimental (pre- and post-tests without follow-up) with non-randomised groups.	Untreated control group	RPWS; BMSQ; FFMQ; CSAI-2; STQ.	A 7-week psychological skills and mindfulness training intervention with sessions focused on attention training, arousal control, breathing exercises, psychological centring, goal-setting, self-talk, imagery, performance under pressure, pre-performance routines, and cognitive restructuring.	Participants who underwent the intervention experienced substantial reductions in their cognitive (*p* = 0.012) and somatic state anxiety (*p* = 0.003). Significant improvements were also seen across many psychological wellbeing domains: positive relations with others (*p* = 0.02), self-confidence (*p* = 0.017), anxiety and worry management (*p* = 0.017), concentration (*p* = 0.002), relaxation (*p* = 0.002), motivation (*p* = 0.002), and growth mindset (*p* = 0.027). These changes were not observed in the control group.
**(9) Other Interventions for Music Performance Anxiety**							
**[6] ([6]), Spain**	Professional wind ensemble members. Total (*n* = 10); Male (*n* = 10).	18–27 *M* = 23	One-arm interventional study (pre- and post-test design with repeated measures and no follow-up).	N/A	STAI; CSAI-2R; heart rate variability indices.	High-intensity interval training (HIIT) involving one session of 2 to 4 bouts of 30 s all-out cycling with 4 min recovery, performed within a 7-day period, after a familiarisation session.	Musicians showed considerable reductions in STAI and CSAI-2R anxiety measures (both *p* < 0.05). Heart rate variability increased post-exercise (*p* < 0.05), indicating improved activity and better stress management pre-performance.
**[20] ([20]), Canada**	University music performance students. Total (*n* = 21); Male (*n* = 9); Female (*n* = 12).	*M* = 23.8	One-arm interventional study (pre- and post-test design without a follow-up), with a scheduled performance post-intervention.	N/A	STAI.	Guided imagery exercise focusing on imagining a successful musical performance. Imagery included visualising images of control, a relaxed stage, a friendly audience, and feelings of confidence. Participants were instructed to practice the exercise daily for a week with a 10 min tape.	Music performance students experienced a significant reduction in mean STAI anxiety levels after participating in the guided imagery programme (*p* < 0.0001).
**[78] ([78]), Taiwan**	3rd-to-6th-grade students in a music programme. Total (*n* = 59); Male (*n* = 26); Female (*n* = 33).	8–12	One-arm repeated-measures study with a time-series design. MPA was measured at various timepoints before the performance: 2 months prior, 1 month prior, 30 min prior, and 5 min prior.	N/A	MPAI-A.	Relaxation breathing training (RBT) intervention where students were guided through abdominal breathing exercises with standardised verbal instructions, accompanied by soft background music. - Two weekly 10 min sessions over 2 months.	RBT was associated with a significant short-term decrease in MPA when conducted immediately (5 min) before a judged performance (*p* < 0.01). However, RBT was not effective from 2 months prior to 30 min before the performance—anxiety levels progressively increased, reaching peak levels 30 min before. RBT provides a useful but limited tool.
**[80] ([80]), United States**	University music students enrolled in piano courses. Total (*n* = 45); Male (*n* = 23); Female (*n* = 22). Three intervention groups: CCR (*n* = 9), CR (*n* = 9), and CCR + CR (*n* = 9). Two control groups: standard treatment (*n* = 9) and waitlist (*n* = 9).	Age not provided	Quasi-experimental study with blocked random assignment to one of five groups (three treatment and two control). Pre- and post-test performances (videoed). No follow-up. No allocation concealment.	Standard treatment control group; waitlist control group	AATS; PPAS; pulse rate; performance assessed by blinded raters using BIA and MPC.	A 6-week programme with one weekly 1 h session: - Cue-controlled relaxation (CCR) group: Progressive muscle relaxation training followed by pairing relaxation with a cue word (“calm”) to be used during performance. - Cognitive restructuring (CR) group: Identifying self-defeating thought patterns, analysing how they create a vicious cycle, and replacing them with positive coping statements during performance. - Combined CCR + CR group: Integration of both techniques, but with reduced time for each component due to the combination.	Both the CCR and CR treatment groups displayed significant main effects: CCR on debilitating anxiety (AATS subscale, *p* < 0.05), musical performance competence (MPC, *p* < 0.001), and the anxiety differential (*p* < 0.004), and CR on behavioural anxiety (BIA, *p* < 0.003). The combined treatment (CCR + CR) was equally beneficial across measures, and did not outperform the individual treatments. Finally, the standard treatment control group did not demonstrate any significant differences compared to the waitlist control group.
**[83] ([83]), Austria**	University violin performance students. Total (*n* = 30); Male (*n* = 4); Female (*n* = 26).	*M* = 23.6	Quasi-experimental with randomised groups but no allocation concealment. Pre- and post-tests, and no follow-up.	Goal-setting control group	K-MPAI; SMPQ; MRF-3.	Pre-performance routines (PPRs) vs. goal-setting intervention over 5 weeks. PPRs included centring techniques, deep breathing, muscle relaxation, and attention techniques with daily practice.	No significant differences for state anxiety or performance quality emerged between the PPR and goal-setting groups. The PPR group experienced a marked increase in self-efficacy compared to the goal-setting group (*p* = 0.049).
**[85] ([85]), United Kingdom**	University students attending music performance classes. Total (*n* = 25); Male (*n* = 4); Female (*n* = 21); Intervention (*n* = 12); Control (*n* = 13).	18–31 *M* = 20.9	Quasi-experimental design. Randomised groups. No allocation concealment. Method of randomisation unclear. Measures were taken on four occasions: at audition, in class pre- and post-treatment, and at the final performance (no follow-up).	Untreated control group	PAI; EPI; MPASS; NMAC; heart rate; performance quality rated by experts.	A 15-session programme of individual lessons in the Alexander Technique. Each lesson lasted approximately 30 min, focusing on posture, learning how to move with less strain, and tension management.	In the intervention group, significant improvements were observed compared to the control group in areas of anxiety (NMAC scores, *p* = 0.04), MPA (MPASS, *p* = 0.05), technical quality (rated by expert judges, *p* = 0.03), and heart rate variance (*p* = 0.02), but only in low-stress conditions. High-stress performance contexts lacked significant changes. Accordingly, these results suggest limited benefits of Alexander Technique training, effects that are context-dependent and require further replication.

Abbreviations: AAQ-2, Acceptance and Action Questionnaire-2; AATS, Achievement Anxiety Test Scale; ACQ, Anxiety Control Questionnaire; AMPS, Appraisal of Music Performer’s Stress; APQ, Autonomic Perception Questionnaire; ASI, Anxiety Sensitivity Index; BAI, Behavioural Anxiety Index; BAFT, Believability in Anxious Feelings and Thoughts; BIA, Behavioural Index of Anxiety; BMSQ, Bull’s Mental Skills Questionnaire; BSI, Brief Symptom Inventory; CBT, Cognitive Behavioural Therapy; CIQ, Cognitive Interference Questionnaire; CSAI-2R, Competitive State Anxiety Inventory-2; DASS-21, Depression, Anxiety, and Stress Scale; DFS-2, Dispositional Flow Scale-2; DSP, Derogatis Stress Profile; EFIM, Flow State Scale for Musical Performers (Spanish acronym); EES, Experiential Shame Scale; EPES, Expectations of Personal Efficacy Scale; EPI, Eysenck Personality Inventory; EPQ-R, Eysenck Personality Questionnaire—Revised Short Scale; ESS, Experiential Shame Scale; FFMQ, Five Facet Mindfulness Questionnaire; FSS, Flow State Scale; FZAQ, Fragebogen zur Auftrittsqualität—Fremdeinschätzung scales of the judging panel; K-MPAI, Kenny Music Performance Anxiety Inventory; LAS, Likert Anxiety Scale; M-PAS, Musicians Performance Anxiety Scale; MBI, Maslach Burnout Inventory; MHC-SF, Mental Health Continuum-Short Form; MPA, Music Performance Anxiety; MPAI-A, Music Performance Anxiety Inventory for Adolescents; MPAQ, Music Performance Anxiety Questionnaire; MPFI, Multidimensional Psychological Flexibility Inventory; MPC, Musical Performance Competence; MPQ, Music Performance Quality Rating Form; MPSS, Appraisal of Music Performer’s Stress; MRF-3, Mental Readiness Form-3; NMAC, Nowlis Mood Adjective Checklist; PAI, Performance Anxiety Inventory; PANAS, Positive and Negative Affect Schedule; PAQ, Performance Anxiety Questionnaire; PARQ, Performance Anxiety Response Questionnaire; PASSS, Performance Anxiety Self-statement Scale; PHLM-S, Philadelphia Mindfulness Scale; POMS, Profile of Mood States; PPAS, Piano Performance Anxiety Scale; PQ, Judge-Rated Performance Quality; PRCP, Personal Report of Confidence as a Performer; PRIME-MD-PHQ, Primary Care Evaluation of Mental Disorders Patient Health Questionnaire; PRMD-Q, Performance-related Musculoskeletal Disorders Questionnaire; PSS, Perceived Stress Scale; RBI, Rational Behaviour Inventory; RPWS, Ryff’s Psychological Wellbeing Scale; SADS, Social Avoidance and Distress Scale; SAI, State Anxiety Inventory; SCS, Social Comparison Scale; SMPQ, Self-Efficacy for Musical Performing Questionnaire; SPA, Signs of Performance Anxiety; SSS, Social Skill Scale; STAI, State-Trait Anxiety Inventory; STAI-S, State-Trait Anxiety Inventory (State Subscale); STAI-T, State-Trait Anxiety Inventory (Trait Subscale); STQ, Self-theory Questionnaire; TAI, Test Anxiety Inventory; VAS, Visual Analogue Scale; VLQ, Valued Living Questionnaire.

**Table 2 behavsci-15-00138-t002:** Condensed synopsis of key results, study limitations, and critical analysis.

Category of Study	Main Results and Study Limitations
**Cognitive Behavioural Therapy Interventions**	CBT effectively reduced MPA and improved performance quality. Benefits were maintained in studies with follow-ups. Large variance in techniques used (e.g., exposure to performances, cognitive restructuring, muscle relaxation) could have influenced outcomes, highlighting a need for more standardised approaches. Moderate design quality (control groups and randomisation), enhancing the reliability of these findings. However, sample sizes were heterogeneous, and follow-ups were short.
**Acceptance and Commitment Therapy Interventions**	ACT enhanced psychological flexibility and defusion from MPA-related thoughts, alleviating MPA. Benefits were sustained in studies with follow-ups. Mixed findings regarding performance quality—one study reported improvement, one had minimal change. Extremely small sample sizes with disproportionate gender ratios that affect the generalisability of the results. No studies used control groups, limiting the ability to draw causal conclusions. Follow-ups were short.
**Music Therapy Interventions**	Inconsistent findings: three papers reported significant changes on validated anxiety scales, while only one specifically demonstrated improvements in MPA using a validated MPA measure, thus raising concerns surrounding whether researchers were consistently measuring MPA, as opposed to general anxiety. Most studies relied on small samples of university students, thus limiting the robustness of their findings. No studies had follow-ups. Three of five studies did not have a control group.
**Yoga and/or Mindfulness Interventions**	Yoga and mindfulness interventions demonstrated steady lowering of MPA alongside favourable outcomes for mindful awareness and dispositional flow. The most substantial benefits were observed in programmes with higher frequency and intensity (e.g., home practice combined with group sessions). Used control groups, large sample sizes, some randomisation, few follow-ups. Prominent issues arose surrounding treatment adherence.
**Virtual Reality Interventions**	VRET consistently reduced MPA by simulating realistic performance scenarios. Performance quality gains were reported, particularly when combined with psychoeducation about anxiety. Small sample sizes. Only one of the two studies had a control group. Neither had follow-ups.
**Hypnotherapy Interventions**	Hypnotherapy produced significant decreases in MPA and state anxiety. Of the one study that had a follow-up, effects were sustained. Although both studies implemented control groups and random assignment, a lack of blinding emerged as a key limitation.
**Physiological and/or Biofeedback Interventions**	Mixed findings: Some improvements to HRV but not to MPA alone. In one study, biofeedback did not produce differential results. Moderate methodological quality, one RCT design. No follow-ups—long-term effects remain under-researched.
**Multimodal Interventions**	Treatments yielded significant alleviations in MPA (maintained at follow-ups). Some benefits to performance quality were observed. Studies had moderately robust designs, including control groups, randomisation, and follow-ups. Despite methodological complexity, interventions were thoughtfully crafted, though the heterogeneity of their techniques made it difficult to isolate which elements were the most effective.
**Other Interventions**	Interventions across studies included HIIT, guided imagery, relaxation breathing training, cue-controlled relaxation, pre-performance routines, and the Alexander Technique. Some promising results in state/trait anxiety and short-term reduction in MPA. No follow-ups. Half of studies had control groups. Moderate sample sizes.

Abbreviations: ACT, acceptance and commitment therapy; CBT, cognitive behavioural therapy; HIIT, high-intensity interval training; MPA, music performance anxiety; RCT, randomised controlled trial; VRET, virtual reality exposure training/therapy.

### 3.4. Findings Categorised by Intervention Type

#### 3.4.1. Cognitive Behavioural Therapy

Six quasi-experimental studies evaluated CBT, five of which reported notable benefits for MPA. [81] ([81]) observed that conservatory music students experienced a significant decline in their MPA following CBT treatment (an effect maintained at 4-month follow-up). In undergraduate students who reported debilitating MPA levels, [56] ([56]) found a marked reduction in trait anxiety and MPA after a programme combining CBT, PMR, and temperature biofeedback training. For adolescents with high MPA, [61] ([61]) observed improvements in students who were adherent to a 7-week CBT programme compared to non-adherent students and controls. [15] ([15]) demonstrated that CBT yielded significant gains for anxiety, performance confidence, and performance quality, while buspirone and placebo medications did not. In a study of music students identified as having extreme MPA by their teachers, [36] ([36]) found that CBT and repeated performance practice were both effective at decreasing MPA compared to waitlist controls at 5-week follow-up, although these differences were not immediately apparent. In the study of [45] ([45]), an anxiety sensitivity reduction intervention significantly decreased state anxiety in musicians between performances, while a CBT-based intervention failed to do so.

#### 3.4.2. Acceptance and Commitment Therapy

Each of the three quasi-experimental papers that investigated the efficacy of ACT documented a significant decline in MPA post-intervention. [16] ([16]) implemented an 8-week ACT programme for undergraduate vocalists, which led to reduced MPA and elevated psychological flexibility, with the effects sustained at 3-month follow-up. In [51]’s ([51]) study, musical theatre students exhibited notable positive changes to their MPA, mindful acceptance, psychological flexibility, defusion with MPA-related thoughts, and lessened performance shame following six weeks of ACT training. Likewise, the effects were maintained at 3-month follow-up. The last paper, a study by [34] ([34]), employed a 12-week ACT intervention in student vocalists. It yielded considerable reductions in MPA and strengthened their psychological flexibility, control of anxiety, cognitive defusion, and performance quality as rated by judges. This progress was conserved at 1- and 3-month follow-ups.

#### 3.4.3. Music Therapy

Music therapy and improvisation-assisted desensitisation were examined across five quasi-experimental studies, but the integrity of the findings remains somewhat muddled. Although three papers reported significant changes on state/trait anxiety scales, only one demonstrated improvement on a specific MPA measure. The first paper, a study by [53] ([53]), assessed overly anxious freelance musicians and discovered that a 12-week group music therapy programme greatly improved their trait anxiety and performance confidence compared to a waitlist condition. A comparative study of two different treatments by [48] ([48]) found that music-assisted PMR combined with imagery produced statistically significant decreases in students’ MPA, while an improvisation-assisted desensitisation group displayed no such change. Nonetheless, both groups experienced marked reductions in state anxiety. A similar study by [47] ([47]) reported declines in student pianists’ state anxiety upon completing a 6-week music therapy improvisation and desensitisation protocol. Effects for trait anxiety and MPA questionnaires were insignificant, however. In a recent paper, [17] ([17]) showed that although group music psychotherapy failed to generate differences in state/trait anxiety post-intervention, students did undergo significant changes to their Likert scale ratings of anxiety. Finally, [68] ([68]) found no significant differences between music-assisted desensitisation, verbal-only desensitisation, and waitlist control groups in terms of physiological MPA measures (heart rate and EMG) or self-reported MPA scales.

#### 3.4.4. Yoga and/or Mindfulness

Five quasi-experimental studies examined yoga and/or mindfulness interventions. Firstly, [13] ([13]) implemented an 8-week meditation programme amongst university music students and reported that, post-intervention, students’ MPA scores dropped significantly relative to waitlist controls. Notably, however, their state anxiety remained unchanged. Shortly thereafter, [50] ([50]) delivered a Zen meditation treatment to conservatory students, wherein a positive correlation emerged between performance quality and MPA in the meditation group, and a negative correlation between the two was observed in the control group. Changes in overall performance quality for both groups were insignificant, however. Following this paper, [76] ([76]) conducted a yoga and meditation programme for conservatory students and found substantial decreases in MPA and trait anxiety following treatment. These improvements were maintained at 7–14-month follow-up. Echoing the previous findings, the largest study amongst the reviewed papers (a study by [46]) reported that 135 young adult musicians who participated in a 6-week yoga programme experienced significant reductions in MPA compared to controls, although state anxiety remained unaltered. [11]’s ([11]) paper, which consolidated data from studies over three years, demonstrated that 8-week yoga programmes significantly elevated dispositional flow and mindfulness in young adult musicians, with these improvements associated with lowered MPA in performance settings relative to a control group.

#### 3.4.5. Virtual Reality

Two quasi-experimental studies focused on virtual reality exposure therapy (VRET). [5] ([5]) found that administering VRET in music students of various age cohorts caused significant improvements in performance quality and MPA compared to untreated controls. Additionally, attenuation of state anxiety was noted for individuals in the VRET group with high pre-test anxiety. For professional and student vocalists, [14] ([14]) demonstrated that reductions across every dimension of MPA (affective, cognitive, relational, and behavioural) emerged after a first exposure session of VRET, but not after a second session.

#### 3.4.6. Hypnotherapy

Both quasi-experimental studies that explored hypnotherapy for students struggling with MPA prompted significant alleviations of their symptoms. [74] ([74]) assessed hypnotherapy compared to a neutral treatment control group and found a significant drop in MPA scores post-intervention and at 6-month follow-up relative to controls. Equally, [9] ([9]) discovered that both cognitive hypnotherapy and EMDR significantly lowered state anxiety compared to untreated controls.

#### 3.4.7. Physiological and/or Biofeedback

Regarding the two studies that evaluated biofeedback interventions, the results were somewhat inconsistent. The first study, a quasi-experimental study by [82] ([82]), employed heart rate variability (HRV) coherence biofeedback training alongside emotional regulation techniques and found significant improvements in students’ combined anxiety scores. However, it should be noted that state anxiety, MPA, and average heart rate did not reach conventional significance thresholds on their own. The second study, an RCT by [86] ([86]), demonstrated that one session of slow breathing exercises, whether combined with biofeedback or not, significantly improved state anxiety and HRV in highly anxious musicians. That said, no differential effects were observed between the biofeedback and non-biofeedback groups, making it unlikely that biofeedback was the causal factor per se.

#### 3.4.8. Multimodal Therapies

Nine studies (eight quasi-experimental and one RCT) assessed multimodal interventions that integrated various practices from the therapeutic approaches discussed above. These programmes routinely featured aspects of CBT, relaxation techniques, breathwork, visualisation, goal-setting, mindfulness, and other psychological skills training (see Table 1 for an in-depth description of each study’s programme). [7] ([7]) evaluated an 8-week multimodal group intervention for secondary school students and found a significant reduction in MPA scores, which persisted at two-month follow-up. Nonetheless, these improvements did not translate to performance quality. [27] ([27]) examined an 8-week programme integrating CBT, ACT, and mindfulness techniques and reported that students experienced long-term attenuation of MPA, which lasted a year. Likewise, when [54] ([54]) delivered a programme to performing students and teachers, which was based on mindfulness, positive psychology, and flow theory, the intervention group exhibited a marked decline in MPA compared to controls. A similar psychological skills and mindfulness programme in [77]’s ([77]) study produced substantial therapeutic gains for university students in areas of cognitive and somatic anxiety, confidence, worry management, concentration, and relaxation.

[72] ([72]) implemented a 14-week multimodal programme specifically designed to train MPA coping strategies and found that students displayed significant improvements in state anxiety and performance quality relative to waitlist comparators. An RCT by [59] ([59]) demonstrated that an educational music training intervention dramatically relieved students’ MPA and perceived stress levels compared to a waitlist control group. These improvements were sustained at 2-week follow-up. In a study by [18] ([18]), a performance skills course displayed significant reductions in MPA and state anxiety among postgraduate music therapy students (in contrast to controls). [31] ([31]) showed how musicians of all ages responded positively to a cognitive restructuring and imagery intervention, alleviating MPA and enhancing judge-rated performance quality, maintained at one-month follow-up. Even so, differences in state anxiety and physiological measures were unchanged. [8] ([8]) studied the impact of three active treatment groups on professional orchestral musicians, all of which yielded similar outcomes, with significant reductions in trait anxiety and performer stress—effects that were maintained at a 2-month follow-up.

#### 3.4.9. Other Interventions

Six quasi-experimental studies did not fit neatly into any category. One study by [6] ([6]) delivered sessions of HIIT exercise to professional musicians and reported considerable reductions in state/trait and competitive anxiety. Similar reductions in state/trait anxiety were observed in university students after completing a guided imagery programme in [20]’s ([20]) study. Another study involving university students (a paper by [85] ([85])) found that the Alexander Technique produced significant improvements in MPA and heart rate compared to a control group, although these outcomes were crucially restricted to low-stress performance contexts. In [78]’s ([78]) study of primary and secondary school students, implementing relaxation breathing was associated with short-term benefits to MPA, but only when conducted immediately before performing. Interestingly, a paper by [83] ([83]) on university violinists and pre-performance routines failed to generate any differences in state anxiety or performance quality compared to a goal-setting group. Finally, [80] ([80]) conducted a 6-week intervention among university students comparing cue-controlled relaxation, cognitive restructuring, and their joint use, finding significant reductions in anxiety across both treatment groups, with no additional benefit from the combined protocol.

## 4. Discussion

### 4.1. Summary and Implications of Results

The present systematic review summarises the available research on interventions for music performance anxiety. A total of 40 studies were extracted, consisting of 2 randomised controlled trials, 38 quasi-experimental papers, and a sum of 1365 participants. The included articles were categorised into nine intervention types: CBT, ACT, music therapy/improvisation and desensitisation, yoga and/or mindfulness, VRET, hypnotherapy, biofeedback, multimodal therapy, and a miscellaneous “other” category. When considering the diversity of the interventions examined and the small participant pool exposed to each intervention type, it becomes apparent that relatively few studies with an appropriate study design to produce causal evidence have been performed thus far. With just two published RCTs—one on biofeedback and another on a multimodal intervention—the overall findings do not allow for conclusive recommendations.

Nevertheless, certain observations can be made. Firstly, it is rather remarkable that multimodal therapies emerged as the most frequently delivered interventions, encompassing 22.5% of the total studies. Such therapies are characterised by their eclectic and holistic nature, and while many programmes are thoughtfully designed with similar core tenets in mind (cognitive restructuring, breathwork, and stress regulation techniques), their actual content, modes of delivery, and even their names differ. Their promising findings may speak to the complexity of the problem—that MPA could have multiple simultaneous causes and, therefore, require an integrated approach—yet it also seems likely that the nascency of the field means that most researchers are interested in experimenting with new programmes rather than confirming the effectiveness of established treatment methods.

CBT surfaced as the second-most examined therapy to alleviate MPA. CBT has been proven to train individuals to acknowledge, examine, and restructure the unhelpful thoughts and behavioural patterns contributing to their anxiety, and it may be beneficial in addressing the irrational beliefs and catastrophic thinking that beget MPA symptoms ([12]). The promising results from the reviewed CBT studies could be substantiated by the wider context that CBT is deemed the gold standard for SAD treatment ([2]). Findings for ACT (a third-wave CBT) were similarly encouraging, which might be attributed to its emphasis on promoting psychological flexibility, an arguably helpful method for targeting the attributes of perfectionism and neuroticism associated with MPA ([79]). However, to prove the effectiveness of either therapy, further investigation in this area remains needed.

Interestingly, music therapy/improvisation and desensitisation interventions and yoga and/or mindfulness programmes were equally tied for third-most utilised. Both techniques are relatively progressive means of treatment, with exercises directing attention to the mind–body connection. While the findings for yoga and meditation were favourable, those for music therapy/improvisation and desensitisation were inconsistent. Again, these therapies demand studies with greater methodological refinement before their efficacy can be confirmed. Other non-traditional approaches like VRET, hypnotherapy, and biofeedback were also identified as potential mitigating therapies. Considering that virtual reality provides musicians with the opportunity to simulate performance environments, confront their avoidance behaviours, and gradually habituate to their anxiety, its probable utility for MPA is somewhat unsurprising ([67]). The results on biofeedback, while thought-provoking, remain mixed and, thus, cannot offer broader conclusions.

### 4.2. Quality of Evidence and Limitations of Included Studies

Despite clear advancements in the development of effective interventions for MPA, certain methodological weaknesses persist, which render it difficult to interpret or compare outcomes across studies. Exercising caution when evaluating the evidence base remains imperative, particularly given the high degree of heterogeneity that characterises the field. Within each intervention type, a vast disparity in programme length, session frequency, techniques implemented, inclusion or exclusion of home practice, and mode of delivery (e.g., group versus individual, online versus in-person sessions) was observed. Such inconsistencies make it challenging to ascertain which specific features of each programme delivered the most value. Moreover, it becomes difficult to ensure that interventions maintain high quality and adhere to their theoretical underpinnings.

Another limitation concerns the variability in the over 60 questionnaire measures used to assess MPA and related outcome variables. While some of these instruments are MPA-specific (e.g., the K-MPAI or the PAI), many are either general anxiety measures (such as the STAI), self-developed, or non-validated tools. This raises questions about the reliability of the chosen measures and whether they consistently capture MPA as opposed to general anxiety. Standardisation of the instruments employed would help to facilitate the precision needed to compare the results effectively. Adding to these critiques, most of the examined studies offered minimal information regarding whether participants’ MPA met pathological thresholds. In fact, only one study required participants to fit the established DSM criteria for SAD ([15]). Another limitation was that only one-quarter of the reviewed studies used ecologically valid performance settings to compare MPA scores pre- and post-intervention. Such performance conditions are integral for recording naturalistic anxiety responses, the lack of which could undermine the validity of the findings. It is also crucial to note that many studies failed to report participants’ fidelity in attending the treatment sessions, a fundamental factor impacting interventions’ outcomes and efficacy.

More broadly, methodological weaknesses emerged concerning limited longitudinal data, with only 32.5% of studies including follow-ups, which generally only lasted two to three months. Accordingly, few conclusions can be made about the long-term sustainability of the interventions’ reported benefits. Further constraints include the fact that many studies recruited a small number of participants with sample demographics limited to young adults and music students, which may not be generalisable to professional musicians or older adults. Additionally, true randomisation has proven challenging in the present research, as participants often volunteered for studies expecting certain treatments. Although both RCTs included in this review implemented true randomisation, the extent to which the quasi-experimental studies followed this standard remains unclear, as most did not specify their method for random assignments. The transparent nature of the reviewed interventions also rendered blinding and concealment of group allocations nearly impossible, as it was typically obvious to participants (and those administering the interventions) whether they were engaging in a yoga class, virtual reality session, or other similar treatment. Weak experimental designs in this regard may have positively biased the findings of the reviewed studies.

### 4.3. Strengths and Limitations of the Current Review

This systematic review constitutes the most recent narrative synthesis of published interventions for music performance anxiety. Compared to previous reviews, the present authors adopted methodically rigorous eligibility criteria, strictly including studies that applied quantitative, pre–post interventional designs and used recognised MPA outcome measures. Earlier reviews routinely evaluated case reports and papers that employed unreliable outcome tools, potentially undermining the conclusiveness of their evidence ([10]; [22]; [26]). Accordingly, this review’s selection strategy enhanced the internal validity of the findings relative to prior reviews. Narrative synthesis enabled a nuanced and clearly organised summary of the research base, one particularly well suited to the heterogeneous interventions and outcomes that comprise the field. It also encouraged a richer interpretation of the findings, accommodating the complexity of the study characteristics and quantitative data that did not fit neatly into a meta-analysis framework.

Nonetheless, several noteworthy limitations restrict the causal impact of our findings. For one, constraints regarding time, funding, and translation resources meant that this review was limited to research published in English or German. The author acknowledges the risk of this decision: excluding foreign-language papers could have led to an underrepresentation of null results, albeit unintentionally ([65]). In much the same way, our exclusion of case reports means that qualitative insights of prospective value were omitted, including a growing body of research into psychodynamic therapy for MPA, which lends itself to a case study design. It is also possible that several relevant papers were inadvertently overlooked, particularly if they were published in lower-impact journals. Equally important to note, the protocol of this systematic review was not registered through PROSPERO, despite an awareness that this would improve the transparency of the project. As an inherent limitation of narrative synthesis, complete objectivity was unattainable given that subjective judgements may have influenced the interpretation of the studies. Moreover, in contrast to meta-analysis, narrative synthesis could not assist in calculating intervention efficacy estimates. A network analysis would have been required to support direct comparisons between the various therapeutic interventions and, ultimately, ascertain the most effective treatment.

### 4.4. Future Directions

By identifying the patterns and methodological weaknesses that underscore the existing literature, this review has the potential to inform clinical decision-making and guide prospective research. One of the primary challenges impacting the field at present is a widespread failure to recognise that statistical significance does not necessarily equate to clinical significance. Researchers have yet to clearly define what level of improvement—such as a specific reduction in K-MPAI scores—equates to clinical value for individuals suffering from MPA. Remedying this gap in knowledge is imperative, as, without a definitive threshold for what is clinically meaningful, it is difficult to determine the appropriate sample sizes needed for randomised controlled trials that rely on power calculations. Naturally, addressing this issue is the first step needed to rectify the overall paucity of randomised controlled trials in the field. According to the best-practice guidelines by the World Federation of Societies of Biological Psychiatry ([29]), a minimum of two positive RCTs are required before a treatment can be recommended with confidence. Importantly, both RCTs should be favourable of the intervention chosen, implement suitable sample sizes, be double-blinded, and use placebo control groups. As such, future research should aim to complete at least two RCTs in this manner for each of the intervention types discussed.

Echoing [37]’s ([37]) reflections, future studies should continue to work towards developing standardised measures for MPA and performance quality, employing these tools consistently across studies, and coming to a universal agreement for which scores on the chosen MPA instrument would indicate an impairment or an adaptation in performance quality. As a related recommendation, documenting potential side effects for each intervention could be a valuable avenue for future studies, and this is a topic largely neglected in current research. Consistent use of follow-ups remains essential to build a clinically sound evidence base, as maintenance and relapse rates are among the fundamental determinants for the effectiveness of anxiety-related interventions ([52]). Multiyear follow-ups in future research could help to ameliorate this issue. A further area of exploration could involve testing two or more therapeutic modalities directly against each other, such as comparing CBT and ACT.

Finally, it remains crucial to acknowledge [55]’s ([55]) perspective that “one size does not fit all” with regard to MPA treatment. Taking a holistic treatment approach offers an appropriate recognition of MPA’s differing presentations and often-multifactorial causes. Although, at first glance, MPA can easily be mistaken for somatic anxiety driven by an overactive autonomic nervous system that responds excessively to perceived danger (i.e., performance), its underlying factors are considerably more nuanced. [39] ([39], [40]) has proposed three possible subtypes of MPA that explain qualitative differences in presentation and severity: (i) situational performance anxiety tied to high-pressure performance contexts, such as auditions or solo recitals, without extending to other contexts; (ii) performance anxiety associated with a co-occurring diagnosis of social anxiety disorder; and (iii) performance anxiety accompanied by comorbid panic, depression, and/or unresolved attachment disruption. Accordingly, it would be prudent to research whether certain individuals (such as those with subtypes (i) or (ii)) respond more effectively to cognitive-based therapies such as CBT or ACT as well as somatic relaxation techniques like yoga or biofeedback. In contrast, for individuals with challenges rooted in early trauma and attachment ruptures (subtype iii), research should explore whether they derive greater benefit from in-depth, longer-term talking therapies. An increasing number of case studies exploring psychodynamic therapy illuminate the complex dynamics underpinning prolonged and debilitating MPA, offering initial support for therapies that bring unprocessed emotions to the surface ([40]; [41], [42]; [32]). Psychodynamic therapies may be especially relevant for performing musicians, as music performance becomes closely tied to identity beginning in early childhood, when the mind is formative and relationships with authority figures are ever-evolving ([55]; [71]). Studies that investigate the efficacy of each intervention type whilst distinguishing amongst these three MPA subtypes could provide critical insights into the possibility of tailored interventions that meet the unique needs of each musician.

## 5. Conclusions

The present systematic review summarises the landscape of therapeutic interventions for music performance anxiety. Despite the aforementioned limitations, the obtained results showcase the possibilities of traditional psychotherapies like CBT or ACT in reducing MPA-related symptoms. Innovative approaches like yoga/meditation, virtual reality, music therapy, integrative multimodal programmes, biofeedback, and hypnotherapy also display considerable promise. However, the shortage of RCTs and the heterogeneity of existing studies preclude any definitive conclusions or recommendations. Prospective research should prioritise rigorous, double-blind, placebo-controlled trials and develop a universally accepted MPA metric that can measure clinically significant changes reliably before any strong endorsements can be made.

## Data Availability

Not applicable.

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
