# Peer review of "Therapeutic Interventions for Music Performance Anxiety: A Systematic Review and Narrative Synthesis"

_behavsci, 2025, doi:10.3390/bs15020138_

Round 1

Reviewer 1 Report

Comments and Suggestions for Authors

This paper is an impressive systematic narrative review of treatments for MPA. It is very well structured and written. It certainly deserves consideration for publication and it will meet that standard following a few corrections and edits.

Introduction

Interestingly, although it is intuitive to deduce that high MPA risks performance quality, that has not been found to be the case. In the event of high MPA, performance will only falter if there has also been inadequate preparation and poor choice of repertoire beyond the level of task mastery achieved on one's instrument. MPA affects a musician's QOL and livelihood because performance becomes so aversive, that despite "getting through it", it can no longer be tolerated. Severe MPA usually co-occurs with other mental health conditions, including one or more of the anxiety disorders and depression, and there may be unresolved attachment issues that render the audience frightening and unsupportive which increases MPA severity. All of these features of MPA need to be included in the brief introductory section of your paper. Finally, I do not find the juxtaposition of adaptive/facilitative MPA against maladaptive/debilitating MPA helpful. It implies a uni-linear construct that varies only along the dimension of severity, which is to misrepresent the complexity of the construct.

Methods

I note that case studies were excluded. I understand your reasoning for doing this, but I feel that case studies are valuable preliminary sources of information that can generate hypotheses for later larger experimental/RCT studies.  I would argue they are more useful than studies using quantitative/questionnaire designs but with very small samples. In such studies we are often left with uninterpretable results – means with very large standard deviations that illuminate little but that are often over-interpreted.

There are a growing number of case studies exploring dynamic psychotherapy for severe MPA that illuminate the complex dynamics underpinning prolonged, severe MPA that is observed in even the most skilled professional musicians. They are well worth including in this updated review for the sake of completeness. 

Perhaps consider

Nagel, J. J. (2010). Treatment of music performance anxiety via psychological approaches: A review of selected CBT and psychodynamic literature. Medical problems of performing artists25(4), 141-148.

Hoffman, T. (2019). The psychodynamics of performance anxiety: psychoanalytic psychotherapy in the treatment of social phobia/social anxiety disorder. Journal of Contemporary Psychotherapy49(3), 153-160.

Spahn, C. (2015). Treatment and prevention of music performance anxiety. Progress in brain research217, 129-140.

Kenny, D. T. (2016). Short-term psychodynamic psychotherapy (STPP) for a severely performance anxious musician: a case report. J. Psychol. Psychother6(272), 2161-0487.

Kenny, D. T., Arthey, S., & Abbass, A. (2014). Intensive short-term dynamic psychotherapy for severe music performance anxiety: assessment, process, and outcome of psychotherapy with a professional orchestral musician. Medical Problems of performing artists29(1), 3-7.

Kenny, D. T., Arthey, S., & Abbass, A. (2016). Identifying attachment ruptures underlying severe music performance anxiety in a professional musician undertaking an assessment and trial therapy of Intensive Short-Term Dynamic Psychotherapy (ISTDP). SpringerPlus5, 1-16.

Also (for other treatments)

Candia, V., Kusserow, M., Margulies, O., & Hildebrandt, H. (2023). Repeated stage exposure reduces music performance anxiety. Frontiers in Psychology14, 1146405.

Finch, K., & Moscovitch, D. A. (2016). Imagery-based interventions for music performance anxiety: an integrative review. Medical problems of performing artists31(4), 222-231.

Discussion

It is true that MPA is multifactorial in nature and would therefore respond to carefully selected multi-modal therapies. Most researchers assume that MPA is due to somatic anxiety expressed as an overactive autonomic nervous system that responds with excessive arousal in the face of a "dangerous" situation (in this case, performance). However, other forms of MPA are primarily cognitively based and would respond more to cognitive therapies compared with e.g., yoga, relaxation etc. Others with MPA have more global personality disturbance rooted in attachment disorders/ruptures that require a psychodynamic therapeutic approach. It would be worth distinguishing among these differing presentations of MPA and their underlying causes and discuss the possible future directions for treatment studies that distinguish among these MPA subtypes.

Author Response

For our point-by-point response see uploaded Word file.

Reviewer 2 Report

Comments and Suggestions for Authors

Dear Authors, I would like to compliment your excellent research work; your review study about interventions on MPA is one of the most comprehensive and rigorous I have read.

The reviewed systematic research aimed to provide an overview and assessment of treatments for music performance anxiety (MPA). The authors followed PRISMA principles and thoroughly searched three databases (PsychInfo, Web of Science, and PubMed) to find quantitative pre-post interventional trials. There were 40 trials with 1,365 individuals in all. The primary interventions investigated were cognitive-behavioral therapy, acceptance and commitment therapy, music therapy, yoga/mindfulness, virtual reality, hypnosis, biofeedback, and multimodal treatment. The results point to the beneficial effects of MPA treatment. Still, they also highlight the shortcomings of the available data, pointing to methodological flaws such as small sample sizes and study heterogeneity.

The results fill the gap of previous review studies, representing a notable advancement in MPA intervention reviews. The contribution is thematically aligned with the journal’s scope. The results are interpreted appropriately, and the conclusions are well supported. The article is well-written, and the analyses are presented effectively. The study is conducted to the highest technical standards.

The research paper advances knowledge on MPA interventions with a wide range of cited references.

My only concern lies with the article's length, which could distract the reader’s focus. Therefore, I would suggest that the authors consider moving Table 1 to an appendix at the end of the paper. I would also like to point out an error in the table numbering, as Table 1 is labeled Table 2.  

Kind regards, the Reviewer

Author Response

(The authors gave the same response as above.)
